# Antioxidant, Anti-Inflammatory and Pro-Differentiative Effects of Chlorogenic Acid on M03-13 Human Oligodendrocyte-like Cells

**DOI:** 10.3390/ijms242316731

**Published:** 2023-11-24

**Authors:** Giuliana La Rosa, Concetta Sozio, Luca Pipicelli, Maddalena Raia, Anna Palmiero, Mariarosaria Santillo, Simona Damiano

**Affiliations:** 1Dipartimento di Medicina Clinica e Chirurgia, Università di Napoli “Federico II”, 80131 Napoli, Italy; giuliana.larosa@unina.it (G.L.R.); sozioimma@gmail.com (C.S.); l.pipicelli@studenti.unina.it (L.P.); anna.palmiero2@studenti.unina.it (A.P.); simona.damiano@unina.it (S.D.); 2Dipartimento di Medicina Molecolare e Biotecnologie Mediche, Università di Napoli “Federico II”, 80131 Napoli, Italy; raia@ceinge.unina.it

**Keywords:** phenolic acid, polyphenols, superoxide ion, reactive oxygen species, NOXs, DUOX2, inflammation, oligodendrocytes, apoptosis, differentiation

## Abstract

Chlorogenic acid (CGA), a polyphenol found mainly in coffee and tea, exerts antioxidant, anti-inflammatory and anti-apoptotic effects at the gastrointestinal level. However, although CGA is known to cross the blood–brain barrier (BBB), its effects on the CNS are still unknown. Oligodendrocytes (OLs), the myelin-forming cells in the CNS, are the main target in demyelinating neuroinflammatory diseases such as multiple sclerosis (MS). We evaluated the antioxidant, anti-inflammatory and anti-apoptotic roles of CGA in M03-13, an immortalized human OL cell line. We found that CGA reduces intracellular superoxide ions, mitochondrial reactive oxygen species (ROS) and NADPH oxidases (NOXs) /dual oxidase 2 (DUOX2) protein levels. The stimulation of M03-13 cells with TNFα activates the nuclear factor kappa-light-chain-enhancer of activated B cell (NF-kB) pathway, leading to an increase in superoxide ion, NOXs/DUOX2 and phosphorylated extracellular regulated protein kinase (pERK) levels. In addition, tumor necrosis factor alpha (TNF-α) stimulation induces caspase 8 activation and the cleavage of poly-ADP-ribose polymerase (PARP). All these TNFα-induced effects are reversed by CGA. Furthermore, CGA induces a blockade of proliferation, driving cells to differentiation, resulting in increased mRNA levels of myelin basic protein (MBP) and proteolipid protein (PLP), which are major markers of mature OLs. Overall, these data suggest that dietary supplementation with this polyphenol could play an important beneficial role in autoimmune neuroinflammatory diseases such as MS.

## 1. Introduction

Diets rich in polyphenols, such as curcumin, chlorogenic acid (CGA), resveratrol and quercetin, are an important source of antioxidants that can be used as an appropriate therapeutic strategy for degenerative diseases [1,2]. In addition to antioxidant properties, numerous studies have attributed a wide range of biological functions to polyphenols, including anti-inflammatory, immunomodulatory and anti-aging activities. Thus, a diet rich in polyphenols protects against chronic diseases by modulating numerous physiological processes, such as cellular redox potential, enzyme activity, cell proliferation and signaling transduction pathways [3,4,5,6].

In both Western and Eastern cultures, tea and coffee beverages, with beneficial effects on health, particularly on the CNS, are widely present in diets.

The neuroprotective effects of coffee could be explained by the action of the different molecules that make up this beverage, such as caffeine [7,8,9,10], theaflavins, catechins, phenylindanes and CGA.

CGA is a member of the group of phenolic acids and is produced by the shikimic acid pathway in plants during aerobic respiration. Specifically, CGA is derived from the condensation of trans-cinnamic acid and quinic acid. A wide variety of natural CGA isomers exist [11], among which the most studied are 3-caffeoylquinic acid and 5-caffeoylquinic acid [12]. 5-caffeoylquinic acid, mainly found in tea, in green coffee beans (76–84% of total CGA) and other plant sources, is the most abundant isomer in the human diet (Figure 1). CGA absorption occurs in the small intestine and colon through the action of the gut microbiota [13,14].

CGA exerts its scavenging activity via a direct hydrogen atom transfer (HAT) reaction from chlorogenic acid to a free radical, or via the formation of a radical intermediate [15,16,17]. In vitro and in vivo studies have demonstrated that the antioxidant, anti-inflammatory and anti-apoptotic [18] effects of CGA in the gastrointestinal tract and liver [19] could be secondary to the reduction in oxidative distress.

Neurodegenerative diseases are characterized by the presence of chronic oxidative stress and dysregulation of the inflammatory response [20]. In addition, during an oxidative state, the secretion of proinflammatory molecules is enhanced, as is apoptosis, and damage-associated molecular pathways are activated. All these factors act together to promote a proinflammatory response [21]. 

Reactive oxygen species (ROS) play important physiological roles in cells [22,23,24,25], but the excessive production of ROS leads to cell damage as they can impair both DNA and protein function and cause the oxidation of cell membrane lipids, leading to cell death.

Oxidative stress also has important functions in the pathogenesis of inflammatory cardiovascular and respiratory, as well as metabolic [26], diseases. In addition, ROS and inflammation are hallmarks of neurodegeneration, contributing to the development of neurodegenerative diseases such as Alzheimer’s disease (AD), Parkinson’s disease (PD), amyotrophic lateral sclerosis (ALS) and multiple sclerosis (MS) [27,28,29,30].

In MS, an autoimmune inflammatory disease characterized by myelin loss in several areas of the CNS, the main cellular target of the autoimmune attack are the oligodendrocytes involved in the production of the myelin sheath that ensure the proper transmission of action potentials in CNS neurons. During CNS development, the oligodendrocytes (OLs) originate from oligodendrocyte progenitor cells (OPCs) that are, in turn, derived from neural progenitor cells (NPCs) [31]. In the adult CNS, large numbers of OPCs retain the ability to migrate, proliferate and differentiate into myelinating OLs. This process is modulated by various factors, including ROS, cytokines, hormones and neurotransmitters [23,32,33]. An important function of OPCs is remyelination, which occurs at the level of white matter lesions in demyelinating diseases [31]. The redox state is crucial for maintaining the balance between the self-renewal and differentiation of OPCs [34]. A redox imbalance in OLs or OPCs can impair their function and myelin sheath formation. Furthermore, it is known that inflammation is associated with oxidative stress with increased ROS levels [35]. The activation of processes leading to neuroinflammation results in the massive and premature arrival of T-cells in the central nervous system, which increases the progression and severity of demyelinating diseases [36].

In recent years, many studies have been conducted to find approaches to reduce oxidative stress and inflammation using supplements to complement the current pharmacological treatments used for MS. It is known that some polyphenols and neuropeptides [37] exert neuroprotective and immunomodulatory actions, thus becoming possible biological tools with great potential for the treatment of immune-mediated disorders of the CNS. CGA is able to cross the BBB [17,38,39]. Moreover, recent pharmacokinetic studies clearly show that chlorogenic acid, and in particular, 5CQA, crosses the BBB and is localized in different nuclei of the CNS [40]. The effects exerted by this polyphenol at CNS levels have not been documented. CGA probably acts directly and/or indirectly on the central nervous system [41].

This in vitro study aims to evaluate whether the antioxidant, anti-inflammatory and anti-apoptotic effects of CGA are extended to the CNS, and in particular, to oligodendrocytes, paving the way for its use as a new dietary supplement capable of reducing immuno-mediated inflammation and favoring OL differentiation. 

## 2. Results

### 2.1. CGA Reduces Superoxide Ions, Mitochondrial ROS and NADPHox Protein Levels

In this study, we used the M03-13 cells (CELLution Biosystem Inc., Toronto, ON, Canada) an immortal human–human hybrid cell line with the phenotypic characteristics of primary oligodendrocytes (OLs). To evaluate CGA’s effects on M03-13 cells, we first performed cell viability assays using growing doses of CGA (10, 25, 100, 250, 500 and 1000 µM). As can be seen in Figure 2, concentrations of CGA up to 250 μM do not impair cell viability.

To evaluate the effects of CGA on superoxide ion levels in oligodendrocytes, the M03-13 cells were incubated with increasing doses of CGA, and then, with a fluorescent probe, dihydroethidium (DHE), that is able to detect cytoplasmic superoxide [42]. 

The fluorescence microscopy images of DHE (Figure 3A) show a reduction in intracellular superoxide ion levels at 25 and 100 μM CGA.

In addition, the effects of CGA on mitochondrial ROS levels were evaluated using the specific mitochondrial fluorescent probe MitoSOX. The fluorescence microscopy images show a significant reduction in mitochondrial ROS levels in the presence of CGA (Figure 3B).

NADPH oxidases are the main membrane ROS-producing enzymes. Specifically, in M03-13 cells, NOX3, NOX5 and DUOX2 isoforms are highly expressed [23,43]. Through Western blot analysis, we evaluated the protein expression levels of NADPH oxidases following CGA treatments (Figure 4A–C). The histograms show a significant reduction in the protein levels of NOX3, NOX5 and DUOX2 as CGA concentrations increase.

### 2.2. CGA Inhibits TNFα-Induced Pro-Inflammatory/Proapoptotic Pathways

Oligodendrocytes are sensitive to cytokines that exert neuro-immunomodulatory functions. Many of these, such as TNFα, generate ROS as second messengers, increasing the activity and the expression levels of NADPH oxidases via TNFR1 signaling. In many cellular models, NOXs-dependent ROS mediate the activation of many TNFα downstream pathways such as the inflammatory/apoptotic and MAPKs pathways.

We also found that in M03-13 cells, TNFα induces superoxide ions via NOXs since in the presence of the NADPHox inhibitor AEBSF, a reduction in TNFα-mediated ROS induction is observed (Figure 5A).

Moreover, NOXs_-_dependent ROS play a role in TNFα-mediated pathway activation in oligodendrocytes. 

One of the key steps in TNFα signaling is the activation of the transcription factor NF-kB (nuclear factor kappa-light-chain-enhancer of activated B cell), which plays an important role in inflammation and innate immunity [44,45]. To highlight NF-kB activation, the protein expression levels of IkBα and NF-kB inhibitor protein were measured. Stimulation with TNFα induces an immediate reduction in cytoplasmic IkBα protein levels followed by an increase in this protein via a negative feedback mechanism protecting against excessive activation of the pro-inflammatory NF-κB pathway [46].

In M03-13 cells, after stimulation with TNFα for 10 min, a decrease in IkBα is observed. Moreover, the incubation of cells with this cytokine induces ERK phosphorylation and PARP cleavage. These effects are reversed in the presence of the NADPHox inhibitor, indicating that a NOXs-dependent increase in superoxide levels mediates the proinflammatory and proapoptotic effects of TNFα in OLs (Figure 5B–D). 

To show the antioxidant role of CGA inn superoxide ions produced via activation of the TNFα proinflammatory pathway, we analyzed intracellular superoxide and NOXs/DUOX2 protein levels in M03-13 cells treated with TNFα at various times in the absence or presence of CGA. The histogram shows a significant increase in intracellular superoxide levels in cells stimulated with TNFα and their reduction in the presence of CGA (Figure 6A). Figure 6A also shows that stimulating M03-13 cells with TNFα in the presence of CGA and the NOXs inhibitor does not show a significant additive inhibitory effect on superoxide levels, suggesting that CGA’s protective effects are exerted mainly through its inhibitory effects on NADPH oxidase activity. In addition, TNFα causes a significant increase in NOX3, NOX5 and DUOX2 protein levels and their reduction in the presence of CGA (Figure 6B–D). 

To evaluate the anti-inflammatory effects of CGA in oligodendrocytes, M03-13 cells were incubated with TNFα at various times in the absence or in the presence of CGA 100 μM, and the activation of NF-kB and ERK was measured via Western blot. These data show that CGA reverses TNFα-induced NF-kB and ERK activation (Figure 7A,B).

Apoptosis occurs through the activation of multiple molecular pathways and effector enzymes, the caspases. In apoptosis, PARP is cleaved by caspases, so increased protein levels of its cleaved form are a good indicator of apoptosis. The activation of caspase 8 leads to a reduction in the protein levels of its inactive form, so in the presence of apoptotic processes, there is an increase in the levels of cleaved PARP and a reduction in the inactive form of caspase 8.

TNFα decreases caspase 8 and increases cleaved PARP protein levels, indicating the activation of the apoptotic pathway by this cytokine. CGA inhibits the pro-apoptotic pathway activated by TNFα (Figure 7C,D). 

### 2.3. CGA Exerts Inhibitory Effects on M03-13 Cell Proliferation, Blocking the Cell Cycle in the G0/G1 Phase

The physiological role of ROS on the differentiation process of many cells of the CNS is described in the literature [47,48]. In particular, ROS are signal molecules involved in the differentiative processes of oligodendrocytes [23].

The OPC microenvironment plays a key role in the regulation of the migration, proliferation and differentiation of these cells, especially inside MS lesions [32]. Much experimental evidence show that cytokines and polyphenols can play an important role in the maturation processes of oligodendrocytes due to their neuro-immunomodulatory abilities.

The antioxidant and anti-inflammatory activity of CGA may modulate the growth and differentiation of OPCs. M03-13 are immortalized cells that move from the proliferative stage (OPCs) to the differentiated stage. One of the methods to induce differentiation is to grow M03-13 cells in the absence of serum. The effects of CGA on cell proliferation were assessed via flow cytofluorimetry using the fluorescent dye CFSE, a lipophilic molecule that can cross cell membranes and diffuse freely within cells. The dye establishes a covalent bond with all the primary amines of intracellular proteins with low toxicity, and the fluorescence levels decrease with cell division. Indeed, the CFSE fluorescence in M03-13 cells is progressively reduced after 24 and 48 h [32]. The addition of CGA for 24 h (25 µM and 100 µM) induces an increase in CFSE fluorescence levels, demonstrating that CGA has an inhibitory effect on oligodendrocyte proliferation (Figure 8A).

In addition, we studied the effects of CGA on the cell cycle and found that, in its presence, there is a block in the G0/G1 phases (Figure 8B). 

### 2.4. Effects of CGA on M03-13 Cell Differentiation 

Our data showing that CGA exerts inhibitory effects on the proliferation of M03-13 cells suggest that CGA may exert pro-differentiative effects. 

The M03-13 cells show some morphological changes during differentiation, the differentiating cells stop proliferating, and multiple processes extend from the cell body. As can be seen in the Figure 9, cells treated for 4 days with a differentiating medium (DMEM no-FBS + PMA) present a more elongated and star-shaped shape, compared to cells cultured in complete growth medium. In addition, cells incubated in complete medium in the presence of 100 µM CGA present a morphology like that of differentiated cells, suggesting that CGA plays an important role in differentiation processes.

To further evaluate the effects of CGA on cell differentiation, the expression levels of differentiation markers were measured in M03-13 cells grown in complete cell medium for 4 days (4d) in the presence or absence of CGA 25 and 100 µM (Figure 10A,B). Myelin basic protein (MBP) and proteolipid protein (PLP) are markers that characterize mature OLs. RT-PCR experiments show that MBP and PLP mRNA levels increase significantly in the presence of increasing doses of CGA (25, 100 μM) compared to cells grown in complete medium (GROW). Cells differentiated with 100 nM Phorbol-12-Myristate-13-Acetate (PMA) were used as a positive control [23].

## 3. Discussion

The results reported in this study indicate that CGA can reduce basal levels of intracellular superoxide, mitochondrial ROS and NOXs/DUOX2 protein expression levels in M03-13 cells. CGA can indirectly act as an antioxidant by stimulating intrinsic cellular antioxidant systems. In addition, in M03-13 stimulated with TNFα, CGA reverses the pro-oxidant, pro-inflammatory and pro-apoptotic effects of this cytokine.

Oligodendrocytes are highly specialized cells with elevated metabolic demand because they produce many myelin internodes, making them the most vulnerable cells of the central nervous system. Various injuries can alter the functionality of OPCs and contribute to the progression of various neurological disorders [49]. Among the main injuries is the alteration of redox balance caused by the increase in intracellular reactive oxygen/nitrogen species (ROS/RNS) concentrations. Major sources of ROS are the mitochondria that regulate vital physiological processes, including the energy production, ion homeostasis and anti-apoptotic mechanisms of OPCs [50]. 

Many studies on CGA, carried out in vivo and in vitro on cell lines of the gastrointestinal tract and heart, have demonstrated its antioxidant and anti-inflammatory role. Studies on rats have shown that CGA inhibits the NF-κB pathway and suppresses p38 MAPK, alleviating intestinal damage induced by chronic stress [51]. Further studies have confirmed that in cardiomyocytes obtained from rats, CGA reduces chemotherapy-induced oxidative stress and apoptosis, through mechanisms involving Nrf2/HO-1 and dityrosine signaling [52]. Reactive oxygen species induce oxidative stress, also promoting the NF-κB signaling pathway in neurodegenerative diseases [53].

Our data showing that in M03-13 stimulated with TNFα, CGA reverses the pro-oxidant, pro-inflammatory and pro-apoptotic effects of this cytokine suggest a protective role of this polyphenol against neuroinflammation. The bioavailability and bioaccessibility of phenolic compounds are affected by the gut microbiota, limiting their positive effects [54] and, therefore, several studies focus on new strategies to increase its bioavailability, such as the use of nanostructured lipid carriers (NLCs) [55].

It is known that OLs undergo a continuous turnover in the adult CNS [31]. The formation of new OLs takes place from their precursors, OPCs, through a series of steps involving the migration, proliferation and subsequent differentiation of OPCs to the stage of myelinating oligodendrocytes. These processes are also necessary steps in the repair mechanisms of demyelinating lesions in MS [56,57,58,59]. In multiple sclerosis, inflammation of the central nervous system plays a central role in disease progression and axonal damage [60]. In patients with multiple sclerosis, in certain brain lesions caused by the inflammatory process typical of this disease, remyelination fails and lesions continue to expand, damaging ever larger areas of nervous tissue. These lesions contribute to the progressive loss of brain function in the most severe forms of the disease.

The data obtained indicate that CGA exerts inhibitory effects on M03-13 cell proliferation, blocking the cell cycle at the G0/G1 phase. In addition, CGA was observed to exert pro-differentiative effects, as shown by the increase in mRNA levels of MBP and PLP, markers of oligodendrocyte differentiation, in M03-13 cells treated with CGA. High levels of ROS often alter the maturation processes of OPCs, preventing the normal repair of lesions and producing an accumulation of neurological deficits and disability over time. Therefore, the antioxidant and anti-inflammatory activity of CGA could account for its pro-differentiative effects in OPCs (Figure 11).

## 4. Materials and Methods

### 4.1. Cell Cultures

In this study, we used M03-13 cells (CELLution Biosystem Inc., Toronto, ON, Canada). The cell line derived from the fusion of a 6-thioguanine-resistant mutant of a human rhabdomyosarcoma with oligodendrocytes obtained from an adult human brain. The M03-13 cells had phenotypic characteristics of primary OLs and were grown in Dulbecco’s Modified Eagle’s Medium (DMEM), containing 4.5 g/L glucose (GIBCO, Thermo Fisher Scientific, Waltham, MA, USA), supplemented with 10% Fetal Bovine Serum (FBS; GIBCO, Thermo Fisher Scientific, Waltham, MA, USA), 100 U/mL penicillin and 100 μg/mL streptomycin. The M03-13 cells were differentiated in FBS-free DMEM (NO-FBS), or in FBS-free DMEM supplemented with 100 nM of Phorbol-12-Myristate-13-Acetate (PMA); (Sigma-Aldrich, Merck, Darmstadt, Germany) for 4 days, and the differentiation cell medium was replaced every day. The cells were kept in a 5% CO_2_ and 95% air atmosphere at 37 °C.

### 4.2. Cell Viability Assay

The toxicity of CGA treatment on M03-13 cells was tested via trypan blue staining. A total of 2.3 × 10^5^ cells were plated in 35 mm Petri dishes in complete DMEM and subsequently starved for 18 h with 0.2% FBS medium, and were treated with increasing doses of CGA (10, 25, 100, 250, 500 and 100 µM). After trypsinization and washing in PBS, the cells were suspended in diluted trypan blue (1:1 with PBS), and then, immediately counted in a Burker’s chamber. The viable (unstained) and unviable (stained) cells were counted separately. The percentage of viable cells was calculated as follows:Cell Viability (%) = [1 − (number of nonviable cells/total number of cells)] × 100.

### 4.3. Western Blotting Analysis

M03-13 cell lysates were obtained in RIPA buffer (150 mM NaCl, 1% NP40, 0.5% deoxycholate, 50 mM Tris HCl, pH 7.5, 0.1% sodium dodecyl sulphate (SDS) containing 2.5 mM Na-pyrophosphate, 1 mM NaVO4, 1 mM β-glycerophosphate,1 mM NaF, 0.5 mM phenyl-methyl-sulfonyl-fluoride (PMSF), and protease inhibitors (Roche Applied Bioscience Penzberg, Upper Bavaria, Germany). The cells were disrupted via repeated aspiration through a 21-gauge needle kept at 4 °C. Cell lysates were centrifuged at 15,871× *g* for 10 min (minutes) and the pellets were discarded. Thirty micrograms of total proteins were subjected to SDS 10% polyacrylamide gel electrophoresis (SDS-PAGE) under reducing conditions. Then, the proteins were transferred onto a nitrocellulose filter membrane (GEHealthcare, Amersham PI, UK) with a Trans-Blot Cell (Bio-Rad Laboratories, Berkeley, CA, USA) in Transfer buffer (25 mM Tris, 192 mM glycine, 20% methanol). For protein detection, membranes were placed in 5% non-fat milk in tris-buffered saline and 0.1% Tween 20 (TWEEN 20 SLCL767, Sigma-Aldrich, Merck, Darmstadt, Germany) at room temperature for 1 h to block the non-specific binding sites. After this, the filters were incubated with mouse polyclonal antibody against IkBα (sc1643, Santa Cruz Biotechnology, Dallas, TX, USA), or a rabbit polyclonal antibody against pERK (877-678, Cell Signaling Technology, Danvers, MA, USA), NOX3 (orb162045, Biorbyt, Cambridge, UK), NOX5 (orb97073, Biorbyt, Cambridge, UK), DUOX2 (orb29192, Biorbyt, Cambridge, UK), caspase 8 (4790, Cell Signaling Technology, Danvers, MA, USA) and cleaved PARP (11835238001, Roche Applied Bioscience Penzberg, Upper Bavaria, Germany), and then, incubated with a peroxidase-conjugated anti-rabbit or anti-mouse secondary antibody (GEHealthcare, Amersham, UK). Peroxidase activity was detected using an enhanced chemiluminescence (ECL) system (Immobilion Western Chemiluminescent HRP Substrate WBKLS010055, Merk Millipore, Darmstadt, Germany) using ChemiDoc (ChemiDoc XRS^+^ Biorad). The membranes were then stripped and probed with an anti α-Tubulin antibody (T9026, Sigma-Aldrich, St. Louis, MO, USA) or an anti GAPDH antibody (E-AB-40337, Elabscience, Houston, TX, USA) to normalize for sample loading and protein transfer. Protein bands were quantified via densitometry using ImageJ software (version 1.8, National Institutes of Health, Bethesda, MD, USA).

### 4.4. DHE (Dihydroethidium) and MitoSOX^TM^ Red Analysis

Intracellular superoxide levels were determined using the fluorescent probe dihydroxyethidium bromide (DHE) (Molecular Probes^TM^, Thermo Fisher Scientific, Waltham, MA, USA) (excitation/emission 518/605 nm). DHE crosses the cell membrane and, at the cytoplasmic level, interacts with the superoxide anion to form a red fluorescent product (2-hydroxyethidium).

The fluorescent probe MitoSOX (Molecular Probes MitoSOX^TM^ Red mitochondrial superoxide indicator, Thermo Fisher Scientific, Waltham, MA, USA) (excitation/emission 396/610 nm) was used for the detection of mitochondrial ROS. M03-13 cells were grown to semi-confluence in 24 multiwell plates in complete DMEM and subsequently starved for 18 h with 0.2% FBS medium in the absence or presence of CGA (10, 25 and 100 μM). 

For the staining, the cells were washed in PBS buffer, and then, were labeled with the 10 μM DHE probe and with 1 μM MitoSOX in FBS-free culture medium. We used 1 μM, because at 5 μM, the signal is not specific to mitochondria and may cause cellular dysfunction due to mitochondrial overload [61].

Next, the cells were immediately fixed in 3.7% paraformaldehyde for 10 min. The coverslips were washed, first in PBS, and then, in distilled water, and finally put on slides for microscopic examination. The cells were analyzed using a Leica DMi8 microscope. Subsequently, the images were analyzed with ImageJ software version 1.8 according to the protocol used by McCloy et al. [62]. Briefly, a line was drawn around each individual cell to calculate the area and intensity of the emitted fluorescence (integrated density). The reported values were normalized with respect to the field background. The total corrected cellular fluorescence (TCCF) was calculated using the following formula:(TCCF) = integrated density − (selected cell area × mean fluorescence of background readings)

The mean value of TCCF was obtained by analyzing 50 cells per sample from three independent experiments performed in triplicate.

### 4.5. Cell Cycle Analysis

For cell cycle analysis, 230,000 cells were plated in 35 mm Petri dishes. The following day, they were placed in serum-free medium for 18 h, except for growing, to promote cell synchronization. On the third day, they were removed from the serum-free medium, and complete DMEM was added while also performing stimulation with CGA 25 and 100 μM. Subsequently, the samples were trypsinized, washed with PBS and centrifuged for 5 min at 159× *g*. The cell pellets were fixed in cold Et-OH 70%, spun at 376× *g* for 5 min at room temperature and washed in PBS. The cell pellets were resuspended in 200 μL of buffer (Triton 0.1%, RNase A 0.1 mg/mL, propidium iodide 10 mg/mL diluted in PBS) and placed on a bascule for 30 min in the dark. The samples were read using FACSCAN (BD, Franklin Lakes, NJ, USA). Data analysis was conducted using FlowJo^TM^ software (version 10, BD, Franklin Lakes, NJ, USA) with the DEAN-JET-FOX model.

### 4.6. CFSE Assay

The M03-13 grown to semiconfluence in 100 mm Petri dishes at confluence were trypsinized, and 1 × 10^6^ cells were resuspended in complete medium containing CFSE 5 µM; then, cells were incubated for 30 min in an incubator at 37 °C in the dark. Next, the cells were washed in PBS buffer to remove the excess of CFSE and resuspended in complete medium (DMEM) in the presence or in the absence of 25 and 100 µM CGA. After 24 h or 48 h the M03-13 cells were trypsinized, centrifuged for 5 min at 159× *g*, washed and resuspended in PBS buffer; then, the samples were analyzed via flow cytometry using FACSCAN (BD, Franklin Lakes, NJ, USA) and the data analyzed using FlowJo^TM^ software version 10.

### 4.7. Crystal Violet Staining Assay for Cell Morphology Evaluation

2.3 × 10^5^ cells were placed in 35 mm Petri dishes in complete DMEM for 18 h. Subsequently, the cells were incubated in the absence and presence of 100 µM *CGA*, in complete DMEM. The differentiated cells were grown in FBS-free medium. The next day, after removing the culture medium and washing the cells in PBS, 500 µL of Crystal Violet was added to the plate for 10 min. After two washes in H_2_O, images were captured using a Leica DMI1 electron microscope.

### 4.8. RNA Extraction and RT-PCR

Total RNA was extracted using TRI-Reagent according to the protocol provided by the manufacturer (Sigma-Aldrich, St. Louis, MO, USA). Total RNA (1 μg) was retrotranscribed using a SensiFASTcDNA kit (Bioline-Meridiam Bioscience Aurogene, Rome, Italy) for 35 min in reaction volumes of 20 μL. RT-PCR was performed using the CFX-Connect^TM^ Bio-Rad system in 96-well reaction plates and at a final volume of 15 μL containing 7.5 μL of iTaq Universal SYBR Green Supermix (Bio-Rad, Hercules, CA, USA), 0.45 μL of the specific primers and 1 μg of cDNA. The gene-specific primers were designed to selectively amplify *MBP* and *PLP*, and their expression values were normalized using the *18S* gene. The fluorescence of SYBR Green was measured at each extension step. The threshold cycle (Ct) reflects the number of cycles in which the generated fluorescence crosses the arbitrary threshold. The reactions were performed at cycle number 40. The primers used are shown in Table 1.

### 4.9. Statistical Analysis

The data are presented as mean ± SEM. As appropriately indicated, differences between groups were compared using ANOVA followed by Bonferroni’s post hoc test to correct for multiple comparisons. In addition, statistical differences between groups were assessed using Student’s *t*-test for unpaired samples, and differences were considered statistically significant at *p* < 0.05. 

## 5. Conclusions

Good nutrition is the basis of good health, and dietary supplements can be a valuable aid in the treatment of various diseases, including neuroinflammatory diseases. The advantage of using supplements is that they have no side effects that alter the patient’s quality of life, but help reduce inflammation and associated clinical symptoms. 

The data obtained, indicating an inhibitory effect of CGA on oxidative, inflammatory, and pro-apoptotic signaling pathways in OLs, suggest that dietary supplementation with this polyphenol could play an important beneficial role in degenerative neuroinflammatory diseases. In addition, the pro-differentiative effects of CGA on OLs suggest its possible beneficial role in remyelination processes occurring inside MS lesions, limiting neuronal loss and the progression of the disease. However, in vivo studies are necessary to confirm the efficacy of this polyphenol in human neuroinflammatory diseases.

Overall, our data suggest that the use of a new combinatory strategy of this polyphenol with current disease-modifying therapies (DMTs) could represent a new therapeutic approach for improving the neurological symptoms and quality of life of MS patients. 

## Figures and Tables

**Figure 1 ijms-24-16731-f001:**
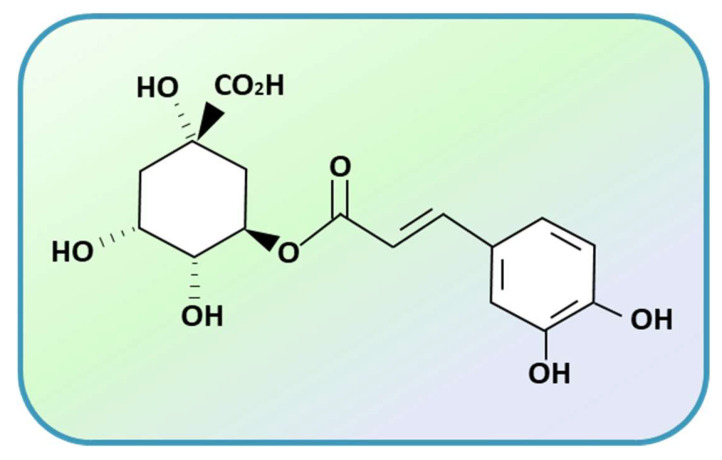
Chemical structure of 5-caffeoylquinic acid (5CQA).

**Figure 2 ijms-24-16731-f002:**
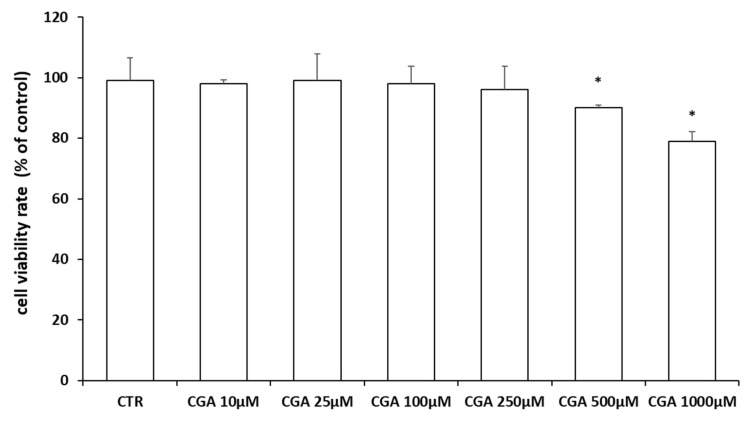
Cell viability according to trypan blue assay in M03-13 cells. The cells were starved for 18 h in 0.2% FBS medium in the absence (CTR) and in the presence of CGA (10, 25, 100, 250, 500 and 1000 µM); the data are reported as percentage variation compared to control. The graph shows the mean ± SEM of the values from three independent experiments. * *p* < 0.05 vs. CTR.

**Figure 3 ijms-24-16731-f003:**
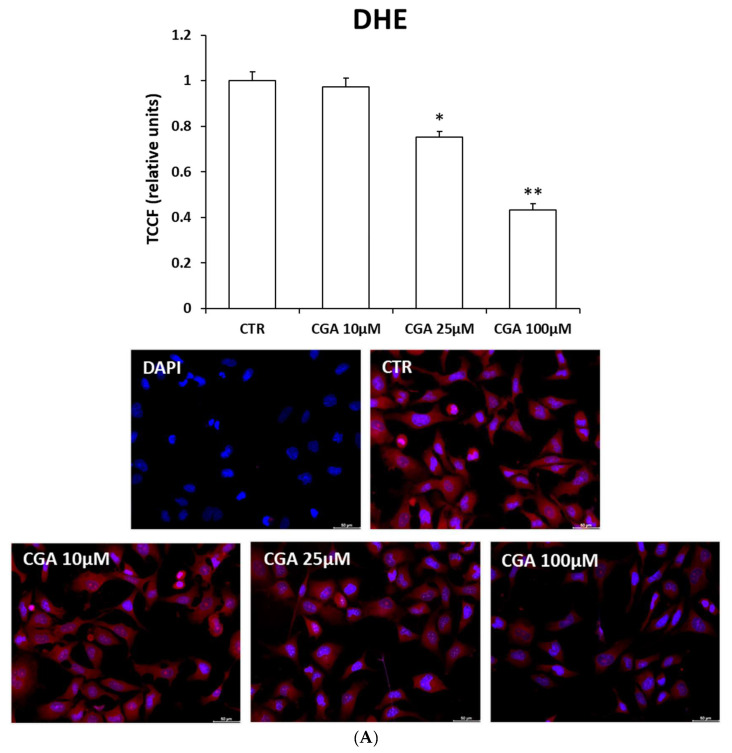
CGA reduces intracellular superoxide ion and mitochondrial ROS levels. Cells were starved for 18 h with 0.2% FBS medium in the absence (CTR) and in the presence of CGA (10, 25, 100 μM). (**A**,**B**) Fluorescence microscopy images of M03-13 cells incubated with 10 μM fluorescent probe DHE (**A**) and 1 μM MitoSOX (**B**) for staining intracellular superoxide ions and mitochondrial ROS, respectively. The histograms (**A**,**B**) show the mean ± SEM total corrected cellular fluorescence (TCCF) values, obtained via quantitative analysis of 50 cells for each sample from three independent experiments performed in triplicate. * *p* ≤ 0.05 vs. CTR. ** *p* ≤ 0.001 vs. CTR. Scale bar is 50 µm.

**Figure 4 ijms-24-16731-f004:**
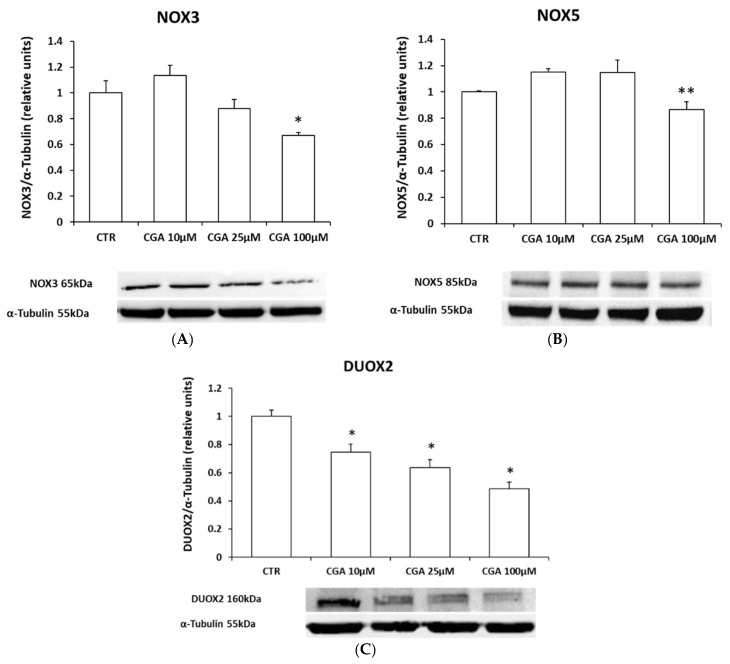
CGA reduces NADPH oxidase levels. Western blotting analysis for NOX3 (**A**), NOX5 (**B**) and DUOX2; (**C**) protein levels in M03-13 cells incubated with 0.2% FBS medium for 18 h in the absence (CTR) and in the presence of CGA (10, 25, 100 μM). The histograms show the values (means ± SEM) relative to the control (CTR), obtained via densitometric analysis of protein bands normalized to α-Tubulin of three independent experiments. * *p* ≤ 0.05 vs. CTR; ** *p* ≤ 0.001 vs. CTR.

**Figure 5 ijms-24-16731-f005:**
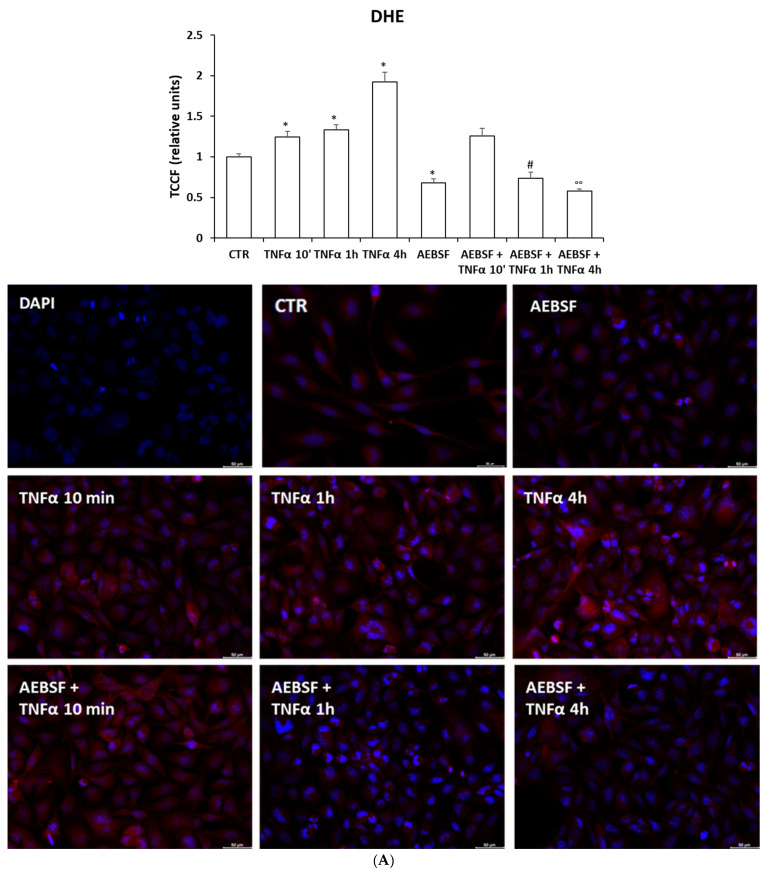
NADPHox-dependent ROS are involved in TNFα-induced pro-inflammatory/proapoptotic pathways. M03-13 were incubated with 0.2% FBS medium for 18 h with 10 µM TNFα in the absence and in the presence of 10 µM AEBSF, and then, incubated with 10 μM of the superoxide probe DHE (**A**). Intracellular superoxide ion levels were measured via fluorometric analysis. The graph shows the mean ± SEM values from three independent experiments (**A**), scale bar is 50 µm. Western blot analysis for IkB⍺ (**B**), pERK (**C**) and the cleaved form of PARP; (**D**) protein levels in M03-13 cells treated with 10 µM TNFα in the presence and absence of 10 µM AEBSF. The histograms show the values (means ± SEM) relative to the control (CTR), obtained via densitometric analysis of protein bands normalized to α-Tubulin of three independent experiments. * *p* ≤ 0.05 vs. CTR; ** *p* ≤ 0.001 vs. CTR; § *p* ≤ 0.05 vs. TNFα 10′; # *p* ≤ 0.05 vs. TNFα 1 h; ## *p* ≤ 0.001 vs. TNFα 1 h; ° *p* ≤ 0.05 vs. TNFα 4 h; °° *p* ≤ 0.001 vs. TNFα 4 h.

**Figure 6 ijms-24-16731-f006:**
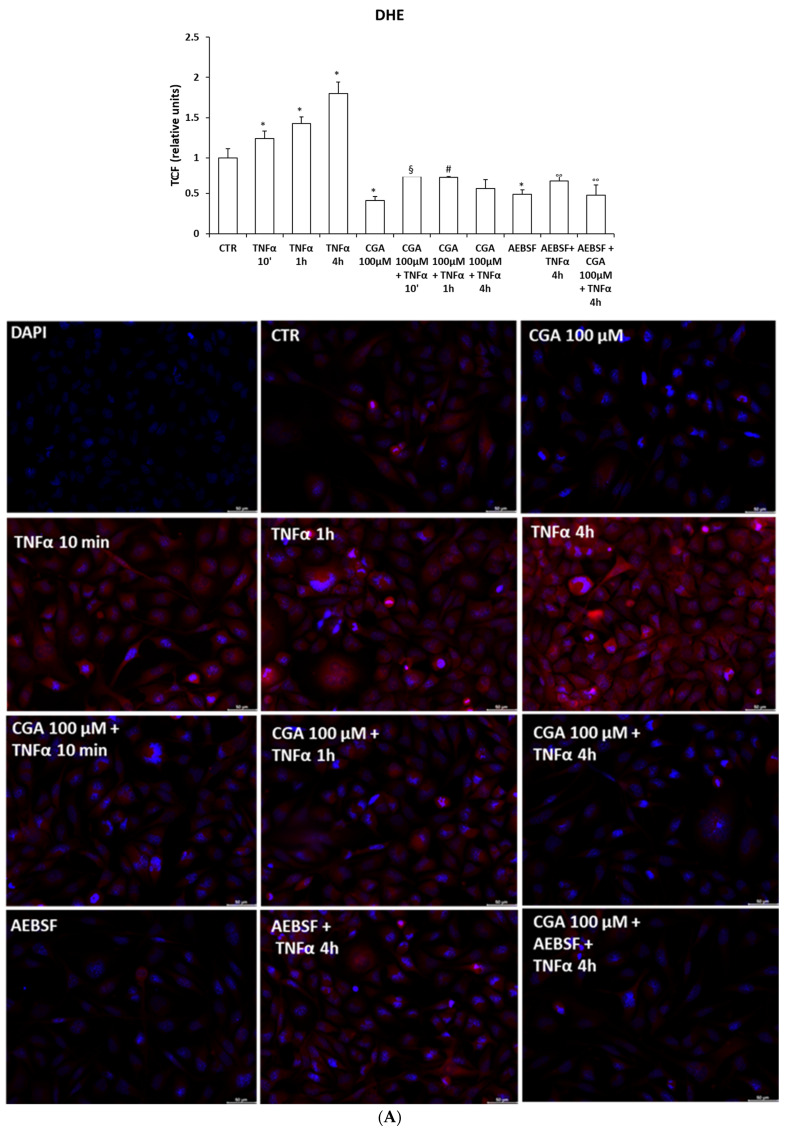
TNF⍺ induces increase in intracellular superoxide levels, measured via DHE fluorescence (**A**). M03-13 cells were preincubated with CGA 100 µM for 18 h and subsequently stimulated with TNFα 10 µM at increasing times (10 min, 1 h and 4 h) in the absence and in the presence of 10 µM AEBSF; then, the cells were incubated with 10 μM DHE and, superoxide levels were measured via fluorometric analysis (scale bar is 50 µm). The graph shows the mean ± SEM values from three independent experiments. Western blot analysis for NOX3 (**B**), NOX5 (**C**) and DUOX2 (**D**). M03-13 cells were incubated with 10 µM TNFα at increasing times (10 min, 1 h and 4 h) in the absence and presence of 100 µM CGA. The histograms show the values (means ± SEM) relative to the control (CTR), obtained via densitometric analysis of protein bands normalized to α-Tubulin of three independent experiments. * *p* ≤ 0.05 vs. CTR; ** *p* ≤ 0.001 vs. CTR; § *p* ≤ 0.05 vs. TNFα 10′; # *p* ≤ 0.05 vs. TNFα 1 h; ## *p* ≤ 0.001 vs. TNFα 1 h; ° *p* ≤ 0.05 vs. TNFα 4 h; °° *p* ≤ 0.001 vs. TNFα 4 h.

**Figure 7 ijms-24-16731-f007:**
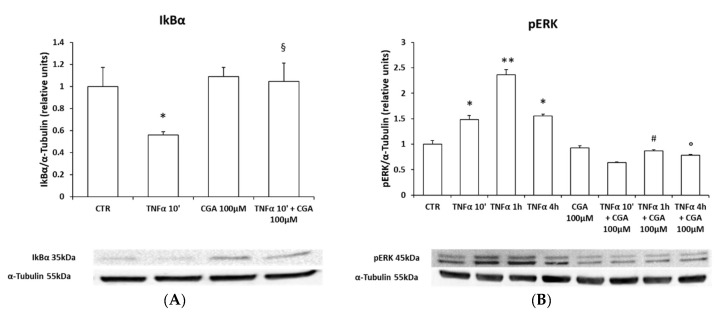
CGA blocks the pro-inflammatory and pro-apoptotic pathways activated by TNF⍺. Western blot analysis for IκBα (**A**), pERK (**B**), caspase 8 (**C**) and cleaved PARP (**D**). M03-13 cells were treated with TNFα 10 µM for 10 min, 1 h, 4 h and 15 h in the absence and/or presence of 100 µM CGA. The histograms show the values (means ± SEM) relative to the control (CTR), obtained via densitometric analysis of protein bands normalized to α-Tubulin of three independent experiments. * *p* ≤ 0.05 vs. CTR; ** *p* ≤ 0.001 vs. CTR; § *p* ≤ 0.05 vs. TNFα 10′; # *p* ≤ 0.05 vs. TNFα 1 h; ° *p* ≤ 0.05 vs. TNFα 4 h; ^ *p* ≤ 0.05 vs. TNFα 15 h.

**Figure 8 ijms-24-16731-f008:**
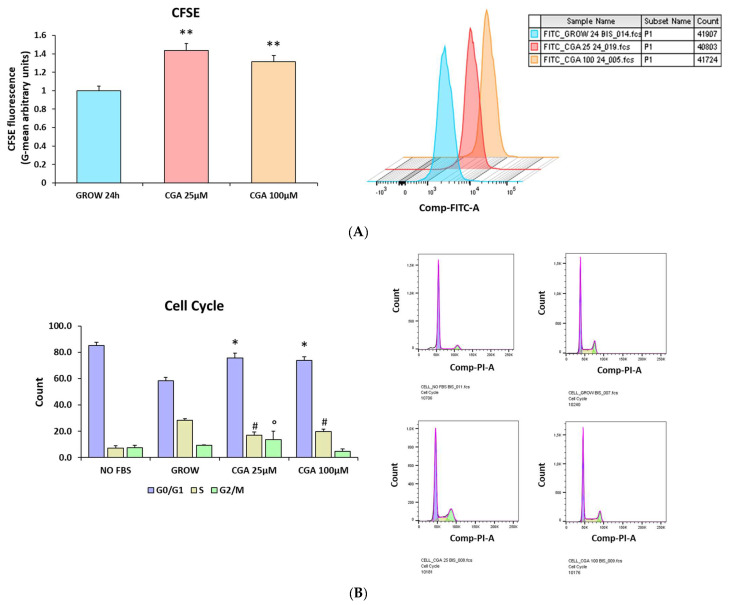
CGA inhibits proliferation and leads to a cell cycle block in M03-13 cells. (**A**) M03-13 cells were incubated for 30 min with CFSE dye and grown in complete medium. A total of 1 × 10^6^ cells per point were analyzed via flow cytofluorimetry after 24 h incubation with the fluorescent dye. Cells were treated with or without CGA 25 µM and 100 µM in the presence of CFSE. The histograms show the means ± SEM of three independent experiments. The graphs of the representative experiments are also shown (**B**). For the cell cycle analysis, the M03-13 cells were treated with CGA for 24 h, and then, treated with propidium iodide (PI) for 30 min; 230.000 cells were analyzed via cytofluorimetry. The histograms show the means ± SEM of three independent experiments, and graphs of the representative experiments are also shown. * *p* ≤ 0.05 vs. GROW (G0/G1); ** *p* ≤ 0.001 vs. GROW 24 h; # *p* ≤ 0.05 vs. GROW (S); ° *p* ≤ 0.05 vs. GROW (G2/M).

**Figure 9 ijms-24-16731-f009:**
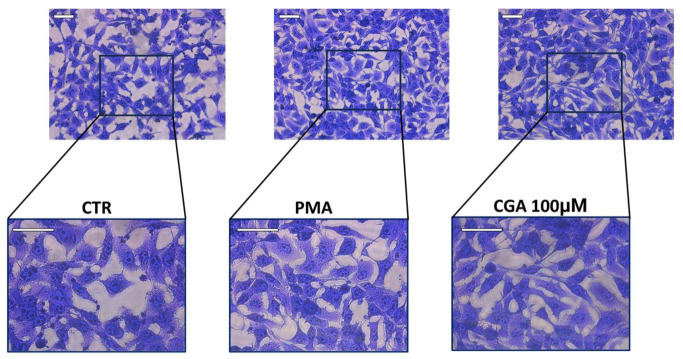
M03-13 cells differentiate in the presence of 100 µM CGA. Crystal Violet staining images of M03-13, to highlight cellular morphological changes, were obtained using a Leica DMI1 microscope. The cells were grown in complete medium for 1 day (CTR) and in serum-free medium in the presence of 100 nM PMA or 100 µM CGA for 4 days (scale bar 100 µm).

**Figure 10 ijms-24-16731-f010:**
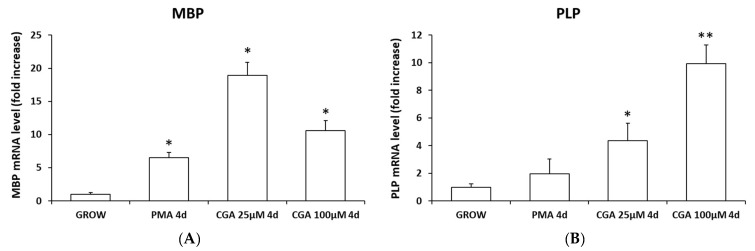
CGA induces an increase in MBP and PLP mRNA levels in M03-13 cells. The M03-13 cells were grown in complete medium for 4 days in the absence (GROW) and in the presence of 25 and 100 µM CGA. PMA 4d indicates differentiated cells grown in serum-free medium in the presence of 100 nM PMA for 4 days. mRNA from treated cells was extracted, and MBP (**A**) and PLP (**B**) mRNA levels were analyzed via real-time PCR. The histograms show the mean ± SEM values of three independent experiments. * *p* ≤ 0.05 vs. GROW; ** *p* ≤ 0.001 vs. GROW.

**Figure 11 ijms-24-16731-f011:**
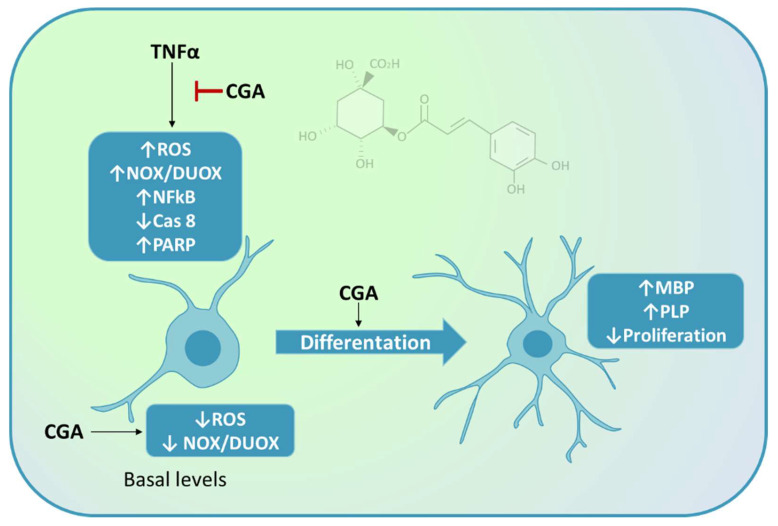
Schematic diagram showing antioxidant, anti-inflammatory and pro-differentiative effects of CGA in OLs. In the presence of CGA, basal levels of ROS and the proinflammatory and pro-apoptotic effects induced by TNFα are reduced. CGA increases mRNA levels of MBP and PLP, major markers of mature OLs. The image was partially created by using BioRender.com.

**Table 1 ijms-24-16731-t001:** The forward and reverse primer of the three genes (*MBP*, *PLP*, *18S*).

Gene	Forward Primer 5′ → 3′	Reverse Primer 3′ → 5′
*MBP*	CGAAGGCCAGAGACCAGGAT	CATGGGTGATCCAGAGCGACT
*PLP*	ATGGAATGCTTTCCCTGGCA	GTAAGTGGCAGCAATCATGA
*18S*	GCGCTACACTGACTGGCTC	CATCCAATCGGTAGTAGCGAC

## Data Availability

The data are contained within the article.

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
