# Peer review of "Antioxidant, Anti-Inflammatory and Pro-Differentiative Effects of Chlorogenic Acid on M03-13 Human Oligodendrocyte-like Cells"

_ijms, 2023, doi:10.3390/ijms242316731_

Round 1

Reviewer 1 Report

Comments and Suggestions for Authors

I think the article is well structured and worked.

However, there are small gaps in the text regarding:

1) Standardize the information regarding the reagents provenance

 2) In the protocols, the spins of the centrifuges are given in rpm; if I want to use the protocol, I would need your centrifuge. Please change.

3) Ascent software- version ?

4) FlowJo - version?

5) BD- I knew it was an American company, not a German one. In Germany there is a large service unit and there is (was ?) the training of BD equipment users'.

Author Response

1) Standardize the information regarding the reagents provenance

We provided to change the text accordingly with the referee’s suggestion in Materials and Methods (lines 400-409; 434-448; 454 and 461; 471-472; 476; 504; 514; 530-531)

 2) In the protocols, the spins of the centrifuges are given in rpm; if I want to use the protocol, I would need your centrifuge. Please change.

We converted rpm in xg in Materials and Methods lines 428, 500, 501, 512)

3) Ascent software- version?

4) FlowJo - version?

We added software version in Materials and Methods lines 468 and 505

5) BD- I knew it was an American company, not a German one. In Germany there is a large service unit and there is (was ?) the training of BD equipment users'.

We provided to change the text as suggested in Materials and Methods line 514

Reviewer 2 Report

Comments and Suggestions for Authors

These paper’s findings are interesting, once neurodegenerative diseases are a public health issue and bioactive compounds have shown to be promising natural agents. Overall, I think the paper can be published after minor revision. The article is robust, and well-written, with complete analyses and original. The suggestions are described below.

 Phenolic compounds can be metabolized during digestion by microbiota, which alters their bioavailability and bioaccessibility. Why did the authors not perform an in vitro digestion to mimetize the gastrointestinal process, before the in vitro assays?

Keywords: I suggest replacing “chlorogenic acid” (it already is in the title) by “phenolic acid”.

Line 96: Replace “polyphelol” by “polyphenol”

Lines 99-102: I suggest evidence that the study was an in vitro model.

Why did the authors use trypan blue staining in the cell viability assay and not MTT?

Why did the authors choose these concentrations of CGA (10, 25, 100, 250, 500, and 1000 μM) for cell viability?

Item 2.9: I do not know this assay. I suggest the authors add its finality.

I suggest the authors evidence the importance of in vivo studies to correlate the findings with MS. The results open avenues to new approaches and research in this area.

Author Response

 Phenolic compounds can be metabolized during digestion by microbiota, which alters their bioavailability and bioaccessibility. Why did the authors not perform an in vitro digestion to mimetize the gastrointestinal process before the in vitro assays?

We agree with the reviewer that bioavailability and bioaccessibility of phenolic compounds are affected by gut microbiota limiting their positive effects but although the topic is very interesting and challenging, unfortunately the study of the effects of microbiota digestion on CGA effects are time consuming experiments and we believe that they are behind the scope of this paper. However, in Discussion (line 370-373) we added the sentence: “Bioavailability and bioaccessibility of phenolic compounds are affected by gut microbiota limiting their positive effects”. Several studies focus on new strategies to increase its bioavailability, such as the use of nanostructured lipid carriers (NLCs).”

We added two references: [54,55] lines 372-373

Keywords: I suggest replacing “chlorogenic acid” (it already is in the title) by “phenolic acid”.

We modified the keywords as suggested

Line 96: Replace “polyphelol” by “polyphenol”

Lines 99-102: I suggest evidence that the study was an in vitro model.

We provided to change the text accordingly with the referee’s suggestion. Lines 96 and 99

Why did the authors use trypan blue staining in the cell viability assay and not MTT?

Cell viability was assessed by thypan blue stainig because the MTT assay estimates cell toxicity through the evaluation of cellular metabolic activity. Because of CGA on cell proliferation and differentiation which are strictly associated with metabolic effects, the MTT assay data could be erroneously interpreted. 

Why did the authors choose these concentrations of CGA (10, 25, 100, 250, 500, and 1000 μM) for cell viability?

We tested concentrations up to 1000 µM to make sure not to use toxic doses. Indeed, the toxic doses of 500 µM and 1000 µM were not used, and the highest dose chosen was 100µM.

Item 2.9: I do not know this assay. I suggest the authors add its finality.

Crystal Violet Staining has been used to evaluate changes in cell morphology associated with OL differentiation. We changed the title of the paragraph in Materials and Methods line 517 with  “Crystal violet staining assay for cell morphology evaluation” to highlight the finality.

We added the finality also in the legend of fig. 9.(line 319)

I suggest the authors evidence the importance of in vivo studies to correlate the findings with MS. The results open avenues to new approaches and research in this area.

We better highlighted that in vivo studies are needed to prove efficacy of polyphenols like CGA in neuroinflammatory diseases, we added in Conclusions (line 554-555) the sentence: “However, in vivo studies are necessary to confirm the efficacy of this polyphenol in humans neuroinflammatory diseases.”

Reviewer 3 Report

Comments and Suggestions for Authors

This is a potentially interesting paper describing antioxidant, anti-inflammatory, and pro-differentiative effects of chlorogenic acid on M03-13 human oligodendrocytes like cells. Authors performed an in vitro studies of cultured immortalized human M03-13 cells stimulated with TNFa and treated with chlorogenic acid. Despite interesting hypothesis and supporting data, there are several issues which requires corrections.

1) Hypothesis.

Authors must understand that there is in the world an antioxidant scavenger of hydroxyl (OH) radical since it instantly react with any organic group. So, delete the invalid statement: "Chlorogenic acid is a scavenger of hydroxyl radical (-OH) [16]".

Please make clear in the paper that polyphenolic compounds can upregulate the expression/activity of cellular antioxidants, therefore, chlorogenic acid can indirectly act as an antioxidant by stimulation of intrinsic cellular antioxidant system.  

2) Methods

Authors used three methods for measurements of reactive oxygen species (ROS): DHE probe, MitoSOX probe and DCFH-DA. SFRBM society and editorial board does not recommend DCFH-DA due to numerous artefacts: Free Radic Biol Med. 2012 Jan 1;52(1):1-6. DCFH-DA does not react/detect any ROS molecule directly rather than indirectly oxidized and it is just a dirty marker of "oxidative stress". Please remove all DCFH-DA data from the paper due to its poor scientific rigor.

Authors must know that MitoSOX at 5uM is not specific to mitochondria and may cause cellular dysfunction due to mitochondrial overload. The reference to "manufacture manual does not help because they do not have the expertise to support this product: Free Radic Biol Med. 2015 Sep; 86: 250–258. Authors must explain the limitations of the protocol they used.

3) Analysis of NADPH oxidase expression

 Authors choose to test the expression of Nox3, Nox5 and DUOX2. Meanwhile, these are the most uncommon isoforms of NADPH oxidase. It must be noted that phagocytic NOX2 is the most critical in oligodendrocytes oxidative stress: J Neuroinflammation. 2013; 10: 155. It is not clear why authors did not test the NOX2 which is the most potent one. Authors are also must explain that NADPH oxidase activity is high regulated and the expression of NOX catalytic isoform does not provide valid information regarding its activity.

4) Please note that stimulation of NFkb pathway leads to upregulation of NADPH oxidase expression/activity. Therefore, the down regulation of NFkb activity by chlorogenic acid could be responsible for diminishing NADPH oxidase expression leading to reduced ROS production, increased cell survival and altering the redox-dependent cell differentiation.

Author Response

1) Hypothesis.

Authors must understand that there is in the world an antioxidant scavenger of hydroxyl (OH) radical since it instantly react with any organic group. So, delete the invalid statement: "Chlorogenic acid is a scavenger of hydroxyl radical (-OH) [16]".

The sentence and the reference have been eliminated.

Please make clear in the paper that polyphenolic compounds can upregulate the expression/activity of cellular antioxidants, therefore, chlorogenic acid can indirectly act as an antioxidant by stimulation of intrinsic cellular antioxidant system.  

We added the sentence suggested by the reviewer in discussion lines 349-350. “CGA can indirectly act as an antioxidant by stimulation of intrinsic cellular antioxidant systems”

2) Methods

Authors used three methods for measurements of reactive oxygen species (ROS): DHE probe, MitoSOX probe and DCFH-DA. SFRBM society and editorial board does not recommend DCFH-DA due to numerous artefacts: Free Radic Biol Med. 2012 Jan 1;52(1):1-6. DCFH-DA does not react/detect any ROS molecule directly rather than indirectly oxidized and it is just a dirty marker of "oxidative stress". Please remove all DCFH-DA data from the paper due to its poor scientific rigor.

To better validate ROS assay we performed new experiments with DHE probe. New figs 5A and 6A

Authors must know that MitoSOX at 5uM is not specific to mitochondria and may cause cellular dysfunction due to mitochondrial overload. The reference to "manufacture manual does not help because they do not have the expertise to support this product: Free Radic Biol Med. 2015 Sep; 86: 250–258. Authors must explain the limitations of the protocol they used.

We thank the reviewer for useful information about the doses of mitosox to use in cell cultures. Analyzing our images, we observed that nuclei only sporadically present probe signal, as shown by the merged images with DAPI, whereas in most cells the signal is barely diffused and mostly spotted indicating mitochondrial staining.  Therefore, we believe that the signal-reducing effect by CGA, may also be partly mediated by an effect on mitochondria. One might hypothesize that the effect of mitosox in uncoupling mitochondria at 5 uM concentration is more evident in neurons and microglia than in oligodendrocytes.

3) Analysis of NADPH oxidase expression

 Authors choose to test the expression of Nox3, Nox5 and DUOX2. Meanwhile, these are the most uncommon isoforms of NADPH oxidase. It must be noted that phagocytic NOX2 is the most critical in oligodendrocytes oxidative stress: J Neuroinflammation. 2013; 10: 155. It is not clear why authors did not test the NOX2 which is the most potent one. Authors are also must explain that NADPH oxidase activity is high regulated and the expression of NOX catalytic isoform does not provide valid information regarding its activity.

We evaluated only protein levels of NOX3, NOX5, and DUOX2 isoforms, because M03-13 cells do not express NOX2 isoform [23, 43]. We added a new reference [43] line 146

The experiments with NOXs inhibitor AEBSF indirectly demonstrate that proinflammatory and proapoptotic effects of TNFα rely on NOXs-dependent ROS also in M03-13 cells demonstrating that TNFα affects NOXs activity.  

4) Please note that stimulation of NFkb pathway leads to upregulation of NADPH oxidase expression/activity. Therefore, the down regulation of NFkb activity by chlorogenic acid could be responsible for diminishing NADPH oxidase expression leading to reduced ROS production, increased cell survival and altering the redox-dependent cell differentiation.

Considering the referee's concerns, we performed new experiments stimulating M03-13 cells with TNFα in presence of CGA and Noxs inhibitor, AEBSF. These substances do not show significant additive inhibitory effects on ROS levels suggesting that CGA protective effects, as well as that of AEBSF, are exerted mainly by its inhibitory effects on NADPH oxidase activity (Fig.6A). We added the sentence in Results (lines 218-222) “Figure 6A also show that stimulating M03-13 cells with TNFα in presence of CGA and Noxs inhibitor it is not observed a significant additive inhibitory effect on ROS levels sug-gesting that CGA protective effects are exerted mainly by its inhibitory effects on NADPH oxidase activity”.

Reviewer 4 Report

Comments and Suggestions for Authors

Authors in their work "Antioxidant, anti-inflammatory, and pro-differentiative effects of chlorogenic acid on M03-13 human oligodendrocytes like cells." investigates an object known to everyone in the world: found in tea and coffee.  In the introduction, the authors have given a good description of the diseases that could potentially be treated with the use of chlorogenic acidThe subject is very topical, it is no secret that we live in a polluted world and efforts to improve this situation are vital. The data presented in this paper is important. The study showed that combining different methods together, it is possible to analyze CGA from different sides, also antioxidant activity is determined. Innovative research methods for comprehensive profiling of CGA have been demonstrated. The work is written really well, all rules are followed, correct statistical evaluation of the data has been carried out. Accepted for publication.

Author Response

We wish to thank the reviewer for her/his positive comments that encourage us to continue our studies in this interesting and exciting field.

Round 2

Reviewer 3 Report

Comments and Suggestions for Authors

This is a revised paper, however, authors  did not respond to several critical problems regarding Methods and poor scientific rigor.

1) Authors used three methods for measurements of reactive oxygen species (ROS): DHE probe, MitoSOX probe and DCFH-DA. SFRBM society and editorial board does not recommend DCFH-DA due to numerous artefacts: Free Radic Biol Med. 2012 Jan 1;52(1):1-6. DCFH-DA does not react/detect any ROS molecule directly rather than indirectly oxidized and it is just a dirty marker of "oxidative stress". Please remove all DCFH-DA data from the paper due to its poor scientific rigor.

2) Authors must know that MitoSOX at 5uM is not specific to mitochondria and may cause cellular dysfunction due to mitochondrial overload. The reference to "manufacture manual does not help because they do not have the expertise to support this product: Free Radic Biol Med. 2015 Sep; 86: 250–258. Authors must explain the limitations of the protocol they used.

3) Authors state that they used "of the ROS sensitive probe s DHE". This is not correct. DHE produces two products: superoxide specific 2-hydroxyethidium and non-specific product ethidium. Therefore, Authors must call DHE a superoxide probe since there is such molecule as "ROS". In addition, all DCF-DA data must be removed since DCF-DA does not react with any specific oxidant. 

Author Response

1) Authors used three methods for measurements of reactive oxygen species (ROS): DHE probe, MitoSOX probe and DCFH-DA. SFRBM society and editorial board does not recommend DCFH-DA due to numerous artefacts: Free Radic Biol Med. 2012 Jan 1;52(1):1-6. DCFH-DA does not react/detect any ROS molecule directly rather than indirectly oxidized and it is just a dirty marker of "oxidative stress". Please remove all DCFH-DA data from the paper due to its poor scientific rigor.

We have removed all DCFH images and data.

2) Authors must know that MitoSOX at 5uM is not specific to mitochondria and may cause cellular dysfunction due to mitochondrial overload. The reference to "manufacture manual does not help because they do not have the expertise to support this product: Free Radic Biol Med. 2015 Sep; 86: 250–258. Authors must explain the limitations of the protocol they used.

We performed new experiments using 1μM mitosox (Fig. 3B) including the reference suggested in methods.

3) Authors state that they used "of the ROS sensitive probe s DHE". This is not correct. DHE produces two products: superoxide specific 2-hydroxyethidium and non-specific product ethidium. Therefore, Authors must call DHE a superoxide probe since there is such molecule as "ROS". In addition, all DCF-DA data must be removed since DCF-DA does not react with any specific oxidant. 

Considering that DHE is a superoxide probe and mitosox a mithocondrial ROS probe we have corrected the text accordingly.

Round 3

Reviewer 3 Report

Comments and Suggestions for Authors

This is a second revision of the manuscript.

Authors have performed additional experiments, revised and updated experimental design and the manuscript in accord with reviewer comments.